# Combining HAIC and Sorafenib as a Salvage Treatment for Patients with Treatment-Failed or Advanced Hepatocellular Carcinoma: A Single-Center Experience

**DOI:** 10.3390/jcm12051887

**Published:** 2023-02-27

**Authors:** Chia-Bang Chen, Chun-Min Chen, Ruo-Han Tzeng, Chen-Te Chou, Pei-Yuan Su, You-Chuen Hsu, Cheng-Da Yang

**Affiliations:** 1Department of Medical Imaging, Changhua Christian Hospital, Changhua 500, Taiwan; 2Big Data Center, Changhua Christian Hospital, Changhua 500, Taiwan; 3Department of Hematology, Changhua Christian Hospital, Changhua 500, Taiwan; 4Division of Gastroenterology & Hepatology, Changhua Christian Hospital, Changhua 500, Taiwan

**Keywords:** hepatocellular carcinoma, hepatic arterial infusion chemotherapy, sorafenib

## Abstract

Background: Hepatic arterial infusion chemotherapy (HAIC) has been proven to be an effective treatment for advanced HCC. In this study, we present our single-center experience of implementing combined sorafenib and HAIC treatment for these patients and compare the treatment benefit with that of sorafenib alone. Methods: This was a retrospective single-center study. Our study included 71 patients who started taking sorafenib between 2019 and 2020 at Changhua Christian Hospital in order to treat advanced HCC or as a salvage treatment after the failure of a previous treatment for HCC. Of these patients, 40 received combined HAIC and sorafenib treatment. The efficacy of sorafenib alone or in combination with HAIC was measured in regard to overall survival and progression-free survival. Multivariate regression analysis was performed to identify factors associated with overall survival and progression-free survival. Results: HAIC combined with sorafenib treatment and sorafenib alone resulted in different outcomes. The combination treatment resulted in a better image response and objective response rate. Moreover, among the patients aged under 65 years old and male patients, the combination therapy resulted in a better progression-free survival than sorafenib alone. A tumor size ≥ 3 cm, AFP > 400, and ascites were associated with a poor progression-free survival among young patients. However, the overall survival of these two groups showed no significant difference. Conclusions: Combined HAIC and sorafenib treatment showed a treatment effect equivalent to that of sorafenib alone as a salvage treatment modality used to treat patients with advanced HCC or with experience of a previously failed treatment.

## 1. Introduction

The poor prognosis of hepatocellular carcinoma (HCC) is a global health concern [1]. A total of 840,000 new cases of liver cancer and 780,000 related deaths were reported in 2018 [2], and it was estimated that more than 1 million people are affected by HCC annually by 2025 [3]. As screening for HCC has become more prevalent in recent years, HCC can be diagnosed in its early stage more frequently, and patients with early HCC can be treated with curative methods such as surgical resection, liver transplantation, or radiofrequency ablation (RFA). Trans-arterial chemoembolization (TACE) or Yttrium-90 radioembolization is the proper treatment for patients who cannot undergo these curative procedures due to their advanced tumor stage, poor liver function, or inadequate liver reserve [4,5]. Alternatively, if the tumor has invaded the major vascular system or if previous treatment has failed, only limited treatment modalities are available, including systemic chemotherapy, targeted therapy, or hepatic arterial infusion chemotherapy (HAIC) [4,5,6,7,8,9,10].

According to several guidelines from Japan, Europe, and the United States, the first-line treatment for patients with advanced or treatment-failed HCC is sorafenib [11]. In addition to sorafenib, HAIC can also be applied. Through the direct delivery of chemotherapy agents into the hepatic tumor-feeding vessels, an increased local chemotherapy drug concentration and reduced systemic toxicity can be achieved [12], and several studies have shown HAIC to be effective in improving the treatment response and survival. 

Traditionally, HAIC has been carried out through the implantation of a chemoport. A modification of HAIC was presented by Tsai [13], who used an angiographic catheter inserted into the left subclavian artery to deliver the chemotherapy drug without a chemoport. The HAIC procedure implemented at our hospital was further modified based on catheter insertion through the left brachial arterial approach, together with the use of sorafenib during the outpatient department follow-up. Previous clinical trials have shown that the addition of HAIC to sorafenib is equally effective as compared to sorafenib therapy alone in terms of overall survival (OS) [14]. 

In this study, we compared the OS and progression-free survival (PFS) of patients with advanced or treatment-failed HCC receiving either a combination of HAIC and sorafenib or sorafenib alone and attempted to identify the possible prognostic factors. 

## 2. Methods

### 2.1. Study Design and Patient Selection

All patients included in the study were affected by advanced or treatment-failed HCC and received sorafenib and/or HAIC treatment between January 2019 and May 2020 at the Changhua Christian Hospital. The inclusion criteria for HAIC were as follows: (1) patients with advanced HCC who were ineligible for surgical resection, RFA, or TACE; (2) patients who had experienced previous treatment for HCC with progressive disease; (3) patients with an adequate liver reserve and serum bilirubin levels < 2 mg/dL; (4) patients without extra-hepatic metastases; (5) patients with ECOG 0 or 1; and (6) patients aged > 20 years old. The exclusion criteria included (1) patients with decompensated hepatic failure and serum total bilirubin levels > 2 mg/dL; (2) patients with extra-hepatic metastases; and (3) patients with an active inflammatory or infections process and a serum white blood cell count > 10,000/μL. 

In total, 75 patients with advanced HCC were enrolled, including 44 patients in the combination group and 31 patients taking sorafenib alone. However, four patients in the combination group were lost to follow-up after the first course of HAIC, and these four patients were excluded. Finally, 71 patients were enrolled in this study, including 40 patients in the combination group and 31 patients receiving sorafenib alone. Data were retrospective collected and analyzed, including age, sex, previous treatments, and laboratory test results including albumin, bilirubin, the Child–Pugh score (CPS), the albumin-bilirubin (ALBI) grade, and alpha-fetoprotein (AFP) before and after procedures. The characteristics of the tumors, including the size, T-stage, vascularity, Barcelona clinic liver cancer (BCLC) stage (0: very early stage; A: early stage; B: intermediate stage; C: Advanced stage; D: any kind of tumor burden), and portal vein tumor thrombus classification according to the liver cancer study group of Japan (PVTT), were recorded after reviewing the CT or MRI and angiography images. The clinical course and 1-year survival rates of all the patients were retrospectively analyzed.

### 2.2. Treatment Protocol

Regarding the HAIC procedure, after the patient was hospitalized and transferred to the angiosuite, sonography of the left subclavian or brachial artery was performed to assess the patency of these vessels. For the 40 patients included in the combination group, the left subclavian arterial was accessed for the 9 starting patients, and the left brachial artery was accessed for the following 31 patients. After puncturing the target artery under sonographic guidance using a micro-puncture set (Cook Medical LLC, Birmingham, UK), a 0.035 inch guidewire and a 4 Fr. Mariner Cobra 1 catheter (Angiodynamics, Latham, NY, USA) were inserted. Angiography of the celiac trunk and the superior mesenteric artery was performed. Embolization of the gastroduodenal artery was routinely performed using metallic coils. In cases of any vascular branch arising from the proper hepatic artery so as to supply the stomach or duodenum, embolization was also performed on these vessels, and if necessary, a microcatheter (1,98Fr. Parkway Soft, Asahi, Japan) was used. However, if these vessels could not be accessed, the catheter tip was located at least 2 cm distal to these vascular branches so as to avoid gastroduodenal injury caused by chemotherapy.

Intra-arterial chemotherapy lasted for 5 days, and the regimen included the following: Cisplatin (10 mg/m^2^) and mitomycin-C (2 mg/m^2^) dissolved in 50 mL of isotonic sodium chloride solution, infused for 20–30 min each and continued for 5 days;100 mg/m^2^ of 5-fluorouracil (5-FU) dissolved in 250 mL of isotonic sodium chloride solution, administered for 24 h using an infusion pump for 5 days;Leucovorin (15 mg/m^2^) every day.

After completion of the 5 days of chemotherapy, 10 mL of lipiodol (Guerbet, Paris, France) was delivered through the catheter for tumoral embolization, followed by the removal of the catheter and manual compressions of the puncture site for hemostasis for 20 min. The patient was discharged after 6 h of observation in the ward. The next course of HAIC was initiated after 4 to 6 weeks, according to the patient’s condition and will. Sorafenib was administered 7 days after discharge and discontinued 7 days before the next HAIC course. A dose reduction or discontinuation of sorafenib was performed according to the clinician’s decision. Computed tomography (CT) or magnetic resonance imaging (MRI) evaluation was performed after one or two cycles of HAIC or every 3–6 months. 

### 2.3. Response and Definitions

The response was defined according to the mRECIST criteria as follows: (1) complete response (CR), indicating the disappearance of any intratumoral arterial enhancement in all the target lesions; (2) partial response (PR), marked by a decrease of at least 30% in the sum of the diameters of the target lesions; (3) progressive disease (PD), marked by an increase of at least 20% in the sum of the diameters of the viable target lesions; and (4) stable disease (SD) or any case that did not qualify for either PR or PD [15]. If CR was achieved, if PD was observed in the following image evaluation, or if the HAIC treatment was not favored by the patients, no further HAIC was arranged, and only sorafenib was prescribed at follow-up in the outpatient department. If SD or PR was achieved, another course of HAIC was arranged and performed for further treatment. 

### 2.4. Follow-Up and Data Collection

Based on a standardized outcome protocol, we conducted a retrospective chart review. The demographics, treatment procedures, and outcomes of patients were collected. All causes of mortality were considered in this study. The primary outcome includes overall survival (OS), which was the primary endpoint and was defined as therapy time from the assignment of therapy to death. From the assignment of treatment to the development of disease progression or death from any cause, progression-free survival (PFS) was calculated. A regular outpatient evaluation was conducted after patients were discharged. A chart review and three-monthly evaluations were conducted in the first year after treatment. 

### 2.5. Statistical Analysis

Statistical analysis was performed using the Statistical Package for the Social Sciences software version 22 (IBM Corporation, Armonk, NY, USA). A chi-square test was performed to demonstrate differences in the baseline characteristics. For continuous variables, the Mann–Whitney U test was used to compare the groups. The 12-month survival rate was estimated using the Kaplan–Meier method with right censoring at the 12-month mark, and the survival outcomes of the groups were compared by the log-rank test. Cox proportional hazards models were used to calculate the OS and PFS. Multivariate Cox regression analysis was performed to determine the prognostic factors for the survival outcomes and to calculate the hazards ratios (HR). The tumor response to treatment was evaluated using the chi-square test. Multivariate analyses of factors that influenced survival were conducted using the Cox proportional hazards model. Statistical significance was defined as a two-sided *p*-value < 0.05.

## 3. Results

### 3.1. Patient Characteristics

The basic characteristics of the enrolled patients are summarized in Table 1. A total of 71 patients were enrolled in our study. These 71 patients had a median age of 66 years (ranging from 42 to 85). Male predominance was observed in both groups. More than 65% of the patients in the combination group had tumors of T-stages 3–4; however, 65% of the sorafenib group had T-stages 1–2. There were also significant differences in the BCLC stages and VP scores of these two groups. The patients in the HAIC group had more advanced HCC than the other group. There were no significant differences in age, prior HCC treatment (TACE/surgery), ascites, the Child–Pugh score, preoperative AFP, albumin, bilirubin, the ALBI score, or the tumor size. 

### 3.2. Responses of the Combination and Sorafenib Groups

The responses of the combination and sorafenib groups are detailed in Table 2. The median PFS tended to be longer in the combination group than the sorafenib group (6 vs. 4 months), but the difference was not statistically significant (*p* = 0.106). The median OS (12 months) was similar in both groups. In the combination group, 14 (35%), 15 (38%), 6 (15%), and 5 (13%) patients exhibited CR, PR, SD, and PD. However, in the sorafenib group, a relatively poor response was noted, with 4 (13%), 6 (19%), 5 (16%), and 16 (52%) patients exhibiting CR, PR, SD, and PD. The objective response rate was significantly higher in the combination group than in the sorafenib group (73% vs. 32%, *p* = 0.001). Considering a reduction in AFP as an indicator of response to treatment, 40% of patients in the combination group showed reductions in AFP, but a reduction in AFP only occurred in 13% of patients in the sorafenib group. Among the patients who had AFP > 20 ng/mL before treatment, more than half in the combination group showed AFP reduction, but this only occurred in a quarter of those in the sorafenib group, indicating a better treatment response in the combination group.

### 3.3. Hazard Rate over One Year

Approximately 93% of progressions and 84–93% of deaths occurred within 9 months after treatment in both groups. The maximum risk of death and progression increased at 6 months, while the risk decreased at 9 months. Compared to the sorafenib group, the combination group had a lower mortality rate (6%) and disease progression rate at 3 months (24%) as well as a lower disease progression rate at 9 months (17%) (Figure 1).

### 3.4. Survival Outcomes

During the first year after treatment, 18 patients in the combination group and 14 patients in the sorafenib group died, while 29 patients in the combination group and 28 patients in the sorafenib group had progressive disease. Kaplan–Meier curves showed that the entire population who received combination treatment had better OS and PFS than the sorafenib group, but no significant statistical difference could be found (*p* = 0.779 in OS and *p* = 0.075 in PFS) (Figure 2). Stratified analysis was performed according to age (Figure 3) and gender (Figure 4). Compared to the sorafenib group, the combination treatment was associated with better PFS among those patients less than 65 years in age (*p* = 0.023 in Figure 3C) and also among the male patients (*p* = 0.045 in Figure 4D). In patients aged < 65, the median PFS was 7.0 months in the combination group, which was better than the 4.0 months observed in the sorafenib group. Moreover, among the male patients, a longer median PFS was observed in the combination group (6 vs. 3 months) compared to the sorafenib group. 

### 3.5. Multivariate Analysis of Tumoral Progression

A subgroup prognostic analysis of all the clinical variables was performed using the Cox proportional hazards model. The results of the multivariate analysis are listed in Table 3 (Model A–B). In Model A, for all patients aged under 65 years, combination treatment (HR = 0.08, 95% CI: 0.02–0.35; *p* = 0.001), preoperative AFP > 400 ng/mL (HR = 5.21, 95% CI: 1.13–23.99; *p* = 0.034), and a tumor size ≥ 3 cm (HR = 6.81, 95% CI: 1.35–34.25; *p* = 0.020) were prognostic predictors of PFS. In Model B, the survival analysis showed that combination treatment (HR = 0.29, 95% CI: 0.11–0.71; *p* = 0.007), an age ≥ 65 (HR = 2.71, 95% CI: 1.33–5.51; *p* = 0.006), and ascites (HR = 4.31, 95% CI: 1.43–13.01; *p* = 0.009) were significant predictors of PFS among male patients. In conclusion, the adjuvant treatment group showed a significant negative relationship with tumor progression in the subgroups analysis.

## 4. Discussion

In this study, we retrospectively evaluated the treatment benefits of combined HAIC-sorafenib and sorafenib alone for patients with advanced HCC or for whom previous treatments failed. Several studies have investigated the effects of sorafenib in combination with other treatments including HAIC [6,7,10,14], TACE [8], chemotherapy [9], and other molecular targeting agents [16]. In Taiwan, sorafenib therapy is the most common treatment modality for these patients due to the fact that its payment is fully covered by national health insurance. Our study revealed the combination treatment results in a better response in the image follow-up and a better objective response rate. The OS and PFS appeared to be better in patients receiving the combination treatment, but there was no significant statistical difference between the two groups. In the multivariate analysis, we observed that male patients who were under 65 years old with tumor sizes < 3 cm and a lower preoperative AFP level (<400 ng/mL) who were treated with combination therapy but were without ascites had a significantly better PFS.

There have been several studies discussing the treatment effects of HIAC or HAIC combined with sorafenib for HCC patients [5,6,7,9,10,11,12,13,14,16,17]. Most showed promising treatment effects of HIAC, with a median OS ranging from 10.1 to 17.1 months compared to sorafenib alone and with a median OS ranging from 6.5 to 10.7 months. Moreover, in patients affected by macroscopic vascular invasion of HCC, HAIC could provide a much better OS than sorafenib alone [10,17]. In our study, patients in the combination group had a more advanced tumor status and more tumors with macrovascular invasion, and combining HAIC with sorafenib as a salvage treatment could provide an OS of approximately 12 months, which is compatible with the results in the literature. 

Recent studies have identified AFP as an important independent risk predictor associated with the pathological grade, progression, and survival risk [18]. Our results showed that patients with AFP > 400 ng/mL had a poorer prognosis than those with AFP < 20 or between 20 and 400 ng/mL. In addition to the serum AFP level, the tumor size is also an indicator of prognosis. A previous study showed that a tumor size ≥ 5 cm was associated with early recurrence and poor overall survival in patients with solitary HCC [19]. Our study showed that hazard ratios for a poor PFS sharply increased in patients with tumor sizes larger than 3 cm. For patients with larger tumors, aggressive treatment and close follow-up for the identification of early recurrence and possible metastases should be considered. 

A gender disparity affecting the prognosis after treatment was noted in our study. It was previously demonstrated that male patients are affected HCC more frequently than females [20,21], which was also noted in our study. Moreover, after curative hepatectomy, a higher early recurrence rate was observed in males compared to females [21], and a longer survival was observed in females with HCC than in males [22]. However, our results showed that male patients had a better PFS after receiving combined HAIC and sorafenib treatment, which is not consistent with results in the literature. More studies and larger study populations are required to identify the potential influence of gender on HCC. 

Patients with ascites are at risk of developing complications and have high mortality. Of the 71 patients in our study, ascites developed in 7 during follow-up. In the multivariant analysis, ascites was considered as a prognostic factor with a hazard ratio of approximately 4.31, indicating a poor prognosis. 

Our study had some limitations. Firstly, the sample size was small since our investigation was a single-center study. Secondly, this study was carried out retrospectively, and selection bias could not be avoided. Additional research is therefore needed, including a validation study with a larger sample size or even a multi-center prospective study. 

## 5. Conclusions

In conclusion, our study revealed that as a salvage treatment modality for patients with advanced HCC or those who underwent a previously failed treatment, combined HAIC and sorafenib treatment, showed equivalent treatment effects to sorafenib alone. Our findings support the prognostic impacts of the baseline tumor size, AFP, and ascites as important factors to consider in trial design. The current analyses also suggest that young male patients aged < 65 years old may benefit more from this combination treatment.

## Figures and Tables

**Figure 1 jcm-12-01887-f001:**
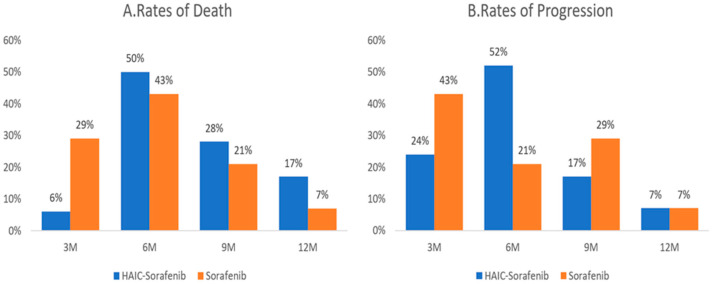
Percentages of death (**A**) and progression (**B**) by treatment strategy at 3 M, 6 M, 9 M, and 12 M (M, months).

**Figure 2 jcm-12-01887-f002:**
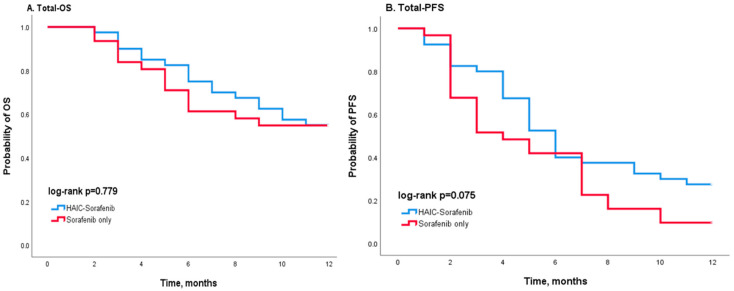
Kaplan–Meier survival analysis according to different treatments (HAIC-sorafenib versus sorafenib alone) for patients with HCC: (**A**) OS and (**B**) PFS. Blue, sorafenib with HAIC; red, sorafenib alone. HAIC, hepatic arterial infusion chemotherapy. Although the survival curves show a better OS and PFS in the combination group than in the sorafenib group, the log-rank test indicates no significant statistical difference between the survival curves.

**Figure 3 jcm-12-01887-f003:**
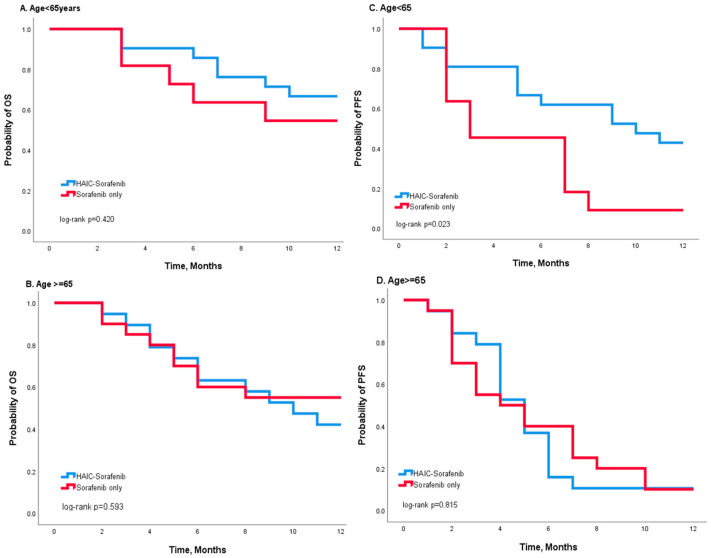
Kaplan–Meier survival analysis according to different treatments (HAIC-sorafenib versus sorafenib alone) for patients of different age groups: (**A**,**B**) OS and (**C**,**D**) PFS. Among those younger than 65 years, the survival curves show a better OS and PFS in the combination group (**A**,**C**), and the log-rank test indicates a significant difference in PFS (*p* = 0.023 in (**C**)). Among those aged 65 and older, the log-rank test shows no significant differences between the survival curves (**B**,**D**).

**Figure 4 jcm-12-01887-f004:**
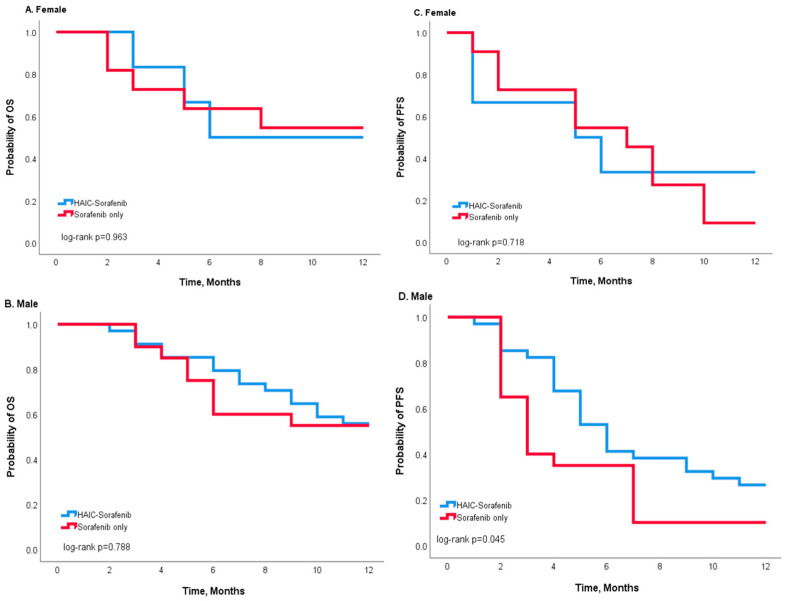
Kaplan–Meier survival analysis according to different treatments (HAIC-sorafenib versus sorafenib alone) for patients of different genders: (**A**,**B**) OS and (**C**,**D**) PFS. The log-rank test indicates no significant differences between the survival curves for females (**A**,**C**). For male patients, the survival curve shows a better OS and PFS (**B**,**D**), and a significant difference was noted in PFS (*p* = 0.045 in (**D**)) according to the log-rank test.

**Table 1 jcm-12-01887-t001:** Patient characteristics.

		HAIC-Sorafenib (*n* = 40)		Sorafenib (*n* = 31)		
		*n*	%	*n*	%	*p*-Value
Age	Aged < 65	21	53%	11	35%	0.153
	Aged ≥ 65	19	48%	20	65%	
Gender	Female	6	15%	11	35%	0.045
	Male	34	85%	20	65%	
Prior treatment	Without	6	15%	4	13%	0.801
	With	34	85%	27	87%	
T-stage	1	0	0%	7	23%	0.002
	2	14	35%	13	42%	
	3–4	26	65%	11	35%	
Ascites	Absence	34	85%	30	97%	0.099
	Presence	6	15%	1	3%	
Child Pugh Score	A5	26	65%	20	65%	0.631
	A6	9	23%	8	26%	
	B7	3	8%	3	10%	
	B8	2	5%	0	0%	
BCLC Stage	A (early stage)	0	0%	7	23%	<0.001
	B (intermediate stage)	23	58%	13	42%	
	C (advanced stage)	17	42%	11	35%	
PVTT stage	0	23	58%	26	84%	0.044
	1–2	6	15%	1	3	
	3	5	12%	1	3	
	4	6	15%	3	10	
Preoperative AFP	<20 (ng/mL)	12	30%	15	48%	0.136
	20–400 (ng/mL)	12	30%	10	32%	
	>400 (ng/mL)	16	40%	6	19%	
Albumin	≤3.5 g/dL	11	28%	13	42%	0.202
	>3.5 g/dL	29	73%	18	58%	
Bilirubin	≥1 mg/dL	13	33%	8	26%	0.540
	<1 mg/dL	27	68%	23	74%	
ALBI score	Mean ± SD	−2.41 ± 0.4		−2.36 ± 0.46		0.643
	Grade 1 (<−2.6)	13	32%	9	29%	
	Grade 2 (−1.39 to −2.6)	27	68%	22	71%	
	Grade 3 (>−1.39)	0	0%	0	0%	
Tumor size	<3 cm	12	30%	14	45%	0.188
	≥3 cm	28	70%	17	55%	
HAIC courses	Mean ± SD	2.03 ± 1.1				
	1	14	35%			
	2	18	45%			
	3	3	7%			
	4	3	7%			
	5	2	6%			

Abbreviation: HAIC, hepatic arterial infusion chemotherapy. BCLC stage, Barcelona clinic liver cancer stage. PVTT, portal vein tumor thrombus. AFP, alpha-fetoprotein. ALBI score, albumin-bilirubin grade score. SD, standard deviation. Footnotes: BCLC stages A, B, and C (Stage A: Early stage. Tumors of any size or up to three tumors less than 3 cm, with well-preserved liver function. Stage B: Intermediate Stage. Tumors in the liver with well-preserved liver function. Stage C: Advanced stage, including invasion of the hepatic blood vessels or extrahepatic spread).

**Table 2 jcm-12-01887-t002:** Efficacy of and Response Rates for Sorafenib and HAIC-Sorafenib Therapy.

		HAIC-Sorafenib (*n* = 40)		Sorafenib (*n* = 31)		
		*n*	%	*n*	%	*p*-Value
Overall survival, months	Median (range)	12.0 (2–12)		12.0 (2–12)		0.594
Time to progression, months	Median (range)	6.0 (1–12)		4.0 (1–12)		0.106
Level of response, No. (%)	Complete response	14	35%	4	13%	0.002
	Partial response	15	38%	6	19%	
	Stable response	6	15%	5	16%	
	Progressive	5	13%	16	52%	
	Response rate	29	73%	10	32%	0.001
AFP response	AFP < 20	11	28%	15	48%	0.033
	Non-AFP reduction group	13	33%	12	39%	
	AFP reduction group	16	40%	4	13%	

Abbreviation: AFP, alpha-fetoprotein.

**Table 3 jcm-12-01887-t003:** Cox regression analyses of DFS in patients aged under 65 (A) and in males (B).

		Factors Associated with Tumor Progression					
		Model A			Model B		
		HR	95% CI	*p*-Value	HR	95% CI	*p*-Value
Sex	Female	1.42	(0.26–7.77)	0.684			
Age	age ≥ 65				2.71	(1.33–5.51)	0.006
Treatment strategy	HAIC-sorafenib	0.08	(0.02–0.35)	0.001	0.29	(0.11–0.71)	0.007
Albumin	alb ≤ 3.5 g/dL	1.23	(0.32–4.74)	0.760	0.99	(0.45–2.17)	0.977
Bilirubin	>1 mg/dL	0.70	(0.19–2.53)	0.586	0.98	(0.45–2.12)	0.962
Preoperative T-stage	Stage 1/2	1.67	(0.38–7.29)	0.497	1.70	(0.72–4.05)	0.227
Preoperative AFP	AFP 20–400 (ng/mL)	1.53	(0.29–8.19)	0.617			0.150
Preoperative AFP	AFP >400 (ng/mL)	5.21	(1.13–23.99)	0.034	1.47	(0.49–4.39)	0.490
Tumor size	≥3 cm	6.81	(1.35–34.25)	0.020	2.75	(0.94–8.05)	0.065
Ascites	With	3.87	(0.31–48.00)	0.292	4.31	(1.43–13.01)	0.009
Prior treatment	With	0.67	(0.09–4.87)	0.692	8.05	(0.75–86.17)	0.085
Child–Pugh Score	Score B7/8	0.26	(0.02–4.20)	0.345	0.78	(0.09–6.68)	0.820

Abbreviations: HR, hazard ratio; CI, confidence interval; AFP, alpha-fetoprotein; HAIC, hepatic arterial infusion chemotherapy.

## Data Availability

Data supporting the findings of this study are available from Changhua Christian Hospital. Restrictions apply to the availability of these data, which were used under license in this study. The data are, however, available from the authors upon request and with permission from Changhua Christian Hospital.

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
