# Peer review of "Combining HAIC and Sorafenib as a Salvage Treatment for Patients with Treatment-Failed or Advanced Hepatocellular Carcinoma: A Single-Center Experience"

_jcm, 2023, doi:10.3390/jcm12051887_

Round 1

Reviewer 1 Report

General comments

This retrospective single-center study by Chen et al. compared the effects including response rate, overall survival (OS) and progression-free survival (PRF) between sorafenib and combined sorafenib with hepatic artery infusion chemotherapy (HAIC) for advanced hepatoma (HCC) or HCC failed to previous treatments. A total of 71 patients including 40 HAIC-Sorafenib and 31 Sorafenib patients was enrolled for analysis. There was significant higher objective tumor response rate (assessed with mRECIST criteria) for HAIC-Sorafenib patients however, with similar OS and PRF. In the subgroup analysis, HAIC-Sorafenib group had better PRF for male with an age less than 65 years old. Although previous randomized study had shown similar OS, this real-world experience demonstrated that HAIC-Sorafenib was beneficial in some selected HCC patients. There are some points need to be clarified.

Major comments

1.     In the baseline clinical characteristics, the authors might provide detailed data including portal vein invasion, BCLC stage, ALBI score, Sorafenib duration and HAIC cycles.

2.     Did HAIC continue after obtaining objective response?

3.     Whether HAIC-Sorafenib patients had higher liver function reserve deterioration rate than Sorafenib group?

4.     Is there better OS for those with objective response rate in HAIC-Sorafenib group? Is there difference in OS between those patients with objective response by HAIC-Sorafenib and Sorafenib?

Author Response

Response to Reviewer1 Comments

Point 1:  In the baseline clinical characteristics, the authors might provide detailed data including portal vein invasion, BCLC stage, ALBI score, Sorafenib duration and HAIC cycles.

Response 1:  

Thanks for your thorough review and your comments. The information including the portal vein tumor thrombus grade (VP grade), BCLC stage, the ALBI score, and the HAIC courses was listed in TABLE 1. The line 89–92 and line 177-181 of the article were also revised.

Point 2: Did HAIC continue after obtaining objective response?

Response 2:

Thanks for your comments. We apologize for the unclear information and have made revision in line 134-138

If CR was achieved, if PD was observed in the following image evaluation, or if the HAIC treatment was not favored by the patients, no further HAIC was arranged and only sorafenib was prescribed at follow-up in the outpatient department. If SD or PR was achieved, another course of HAIC was arranged and performed for further treatment.

Point 3: Whether HAIC-Sorafenib patients had higher liver function reserve deterioration rate than Sorafenib group?

Response 3:

Thanks for this question.

We performed liver function test for every patient, both before and right after the HAIC treatment and also during the OPD follow-up as soon as 1 to 2 weeks after discharge. We had not observed rapid deterioration of hepatic function after HAIC treatment yet. However, we believe that every kind of treatment would damage the liver function and gradually lead to hepatic insufficiency at the end.

Point 4: Is there better OS for those with objective response rate in HAIC-Sorafenib group? Is there difference in OS between those patients with objective response by HAIC-Sorafenib and Sorafenib?

Response 4:

Thanks for the question.

Patients who had objective response after combined HAIC-sorafenib treatment had better OS (305.6 +97.9days) than those with SD (289.6+84.3 days) and those who only had PD (159.4+62.7 days).

In the Sorafenib group, patients achieving objective reponse had OS of 351.6+40.2 days, compared to those with SD (259.4+131.4) and PD (209.9+116.6)

In all patients achieving objective response in these two groups, the sorafenib group showed a little better OS than the HAIC-sorafenib group. However, because only 11 patients(36.7%) had objective response in the sorafenib group, compared to 29 patients (70.7%) with objective response in the HAIC-sorafenib group, further research with bigger study group would be required to compare the treatment effect of these two groups.

Reviewer 2 Report

1. The document presents plagiarism and 28 grammatical errors that must be corrected.

2. You must not present conclusion data in results line 156.

3. Approximately 35% of the bibliography is more than 5 years old, please update.

4. The support in references is poor for the importance of the problem under study, more evidence should be attached regarding the problem and the experience with the treatments used.

5. In conclusion, he comments that there is "In conclusion, our study revealed combining HAIC and sorafenib treatment provides good OS and PFS as sorafenib monotherapy for patients with advanced HCC or those with failure in previous treatment", however, in results he comments that there is no statistical difference. Please harmonize the conclusion with the results.

Author Response

Response to Reviewer 2 Comments

Point 1: The document presents plagiarism and 28 grammatical errors that must be corrected.

Response 1:

Response: The iThecticate system was used before submission. The results of the documentation show a 14% similarity after removing the references. There is only 9% similarity between the text (Introduction, Method, Result, Discussion, and Conclusion), and the similarity between the first sources is less than 2%. Thanks for pointing out the grammatical error part, we will submit it to the professional for English grammar confirmation

Point 2: You must not present conclusion data in results line 156.

Response 2:

Thank you for the comment. The line 156 was revised as the following :

Considering reduction of AFP as an indicator of response after treatment, 40% of patients in the combination group showed reduction of AFP, but reduction of AFP only occurred in 13% of patients in the sorafenib group

Point 3: Approximately 35% of the bibliography is more than 5 years old, please update.

Response 3:

Thank your for your comment. Most of the references before 2017 were all removed and updated references were listed instead.

Point 4: The support in references is poor for the importance of the problem under study, more evidence should be attached regarding the problem and the experience with the treatments used.

Response 4:

Thank you for the comment. Follwing the reviewer’s suggestion, the references were updated and more close to the topic of our article.

Point 5:  In conclusion, he comments that there is "In conclusion, our study revealed combining HAIC and sorafenib treatment provides good OS and PFS as sorafenib monotherapy for patients with advanced HCC or those with failure in previous treatment", however, in results he comments that there is no statistical difference. Please harmonize the conclusion with the results.

Response 5:

Thank you for your comment. We apologize for the unclear information and have revised the conclusion to be consistent with the results:

“In conclusion, our study revealed that as a salvage treatment modality for patients with advanced HCC or those who underwent a previously failed treatment, combined HAIC and sorafenib treatment showed equivalent treatment effects to sorafenib alone. Our findings support the prognostic impacts of the baseline tumor size, AFP, and ascites as important factors to consider in trial design. The current analyses also suggest that young male patients aged < 65 y/o may benefit more from this combination treatment

Response 6:

Thank your for your comment.

A retrospective case-cohort study was conducted for this study. For the purpose of the study, patients with advanced HCC treated with Sorafenib were selected and followed up for one year. We compared the case group receiving HAIC-Sorafenib to the control group who received only Sorafenib treatment during the follow-up period. Multi-factor analysis can then be performed using the Cox model and HR can be calculated.

We apologize for the unclear information and have made revision in Method section in relation to this point: (page 3, line 145-153)

Based on a standardized outcome protocol, we conducted a retrospective chart review. The demographics, treatment procedures, and outcomes of patients were collected.  All causes of mortality were considered in this study. The primary outcome includes overall survival (OS), which was the primary endpoint, and was defined as therapy time from the assignment of therapy to death. From the assignment of treatment to the development of disease progression or death from any cause, progression-free survival (PFS) was calculated. A regular outpatient evaluation was conducted after patients were discharged. A chart review and three-monthly evaluations were conducted in the first year after treatment.

Response 7 :

Thank you for your comment.

We apologize for the misleading part in the discussion, and we have made a revision in the discussion  (page11, line278-286).

There have been several studies discussing the treatment effects of HIAC or HAIC combined with sorafenib for HCC patients. Most showed promising treatment effects of HIAC, with a median OS ranging from 10.1 to 17.1 months, compared to sorafenib alone, with a median OS ranging from 6.5 to 10.7 months. Moreover, in patients affected by macroscopic vascular invasion of HCC, HAIC could provide a much better OS than sorafenib alone. In our study, patients in the combination group had a more advanced tumor status and more tumors with macrovascular invasion, and combining HAIC with sorafenib as a salvage treatment could provide an OS of approximately 12 months, which is compatible with the results in the literature.

Reviewer 3 Report

Manuscript: jcm-2185586

This paper by Chen et al, describes that their data showed combined hepatic arterial infusion chemotherapy (HAIC) and sorafenib increase overall progression-free survival, though several data for overall survival and progression-free survival are not statistically changed. I must mention that the treatment type is not so novel (many studies are already published) and have several limitations including the sample size. However, I appreciate that the authors tried their best to study this challenging research using a single-center data. I have very few minor comments.

1.     In abstract section, the number of patients is 41 receiving combined treatment, but shown 40 in other sections of the paper.

2.     Define mRECIST in line 119.

3.     In line 104, mg/m2 should be mg/m2

4.     Explain statistical analysis in figure legends.

Author Response

Response to Reviewer 3 Comments

Point 1: In abstract section, the number of patients is 41 receiving combined treatment, but shown 40 in other sections of the paper.

Response 1:

Thanks for thorough and careful review, and kind comments . The mistake was corrected.

Point 2: Define mRECIST in line 119.

Response 2:

Thanks for the comment. The brief description of the mRECIST for HCC was discribed in line 129-134, as the following sentences :

The response was defined according to the mRECIST criteria, as follows: 1. complete response (CR), indicating the disappearance of any intratumoral arterial enhancement in all the target lesions, 2. partial response (PR), marked by a decrease of at least 30% in the sum of the diameters of the target lesions, 3. Progressive disease (PD), marked by an increase of at least 20% in the sum of the diameters of the viable target lesions, and 4. stable disease (SD), or any case that did not qualify for either PR or PD

Point 3:  In line 104, mg/m2 should be mg/m2

Response 3:

Thanks for the careful review. The mistake was corrected.

Point 4:  Explain statistical analysis in figure legends.

Response 4:

Thanks for your comment. We apologize for the unclear information and have added Figure legends in relation to this point:

Figure Legends

Figure 2. Kaplan–Meier survival analysis according to different treatments (HAIC-Sorafenib versus Sorafenib alone) for patients with HCC: (A) OS and (B) PFS. Blue, Sorafenib with HAIC; red, Sorafenib alone. HAIC, hepatic arterial infusion chemotherapy. Although the survival curves show a better OS and PFS in the combination group than in the sorafenib group, the log-rank test indicates no significant statistical difference between the survival curves

Figure 3. Kaplan–Meier survival analysis according to different treatments (HAIC-Sorafenib versus Sorafenib alone) for patients of different age groups: (A, B) OS and (C, D) PFS. Among those younger than 65 years, the survival curves show a better OS and PFS in the combination group (A, C), and the log-rank test indicates a significant difference in PFS (p=.023 in C). Among those aged 65 and older, the log-rank test shows no significant differences between the survival curves (B, D).

Figure 4. Kaplan–Meier survival analysis according to different treatments (HAIC-Sorafenib versus Sorafenib alone) for patients of different genders: (A, B) OS and (C, D) PFS. The log-rank test indicates no significant differences between the survival curves for females (A, C). For male patients, the survival curve shows a better OS and PFS (B, D), and a significant difference was noted in PFS (p=.045 in D), according to the log-rank test.

Round 2

Reviewer 2 Report

The paper gratly improved, so I can suggest its publication.